# A Deep Architecture for Matching Short Texts

**Zhengdong Lu**
Noah's Ark Lab
Huawei Technologies Co. Ltd.
Sha Tin, Hong Kong
Lu.Zhengdong@huawei.com

**Hang Li**
Noah's Ark Lab
Huawei Technologies Co. Ltd.
Sha Tin, Hong Kong
HangLi.HL@huawei.com

## Abstract

Many machine learning problems can be interpreted as learning for matching two types of objects (e.g., images and captions, users and products, queries and documents, etc.). The matching level of two objects is usually measured as the inner product in a certain feature space, while the modeling effort focuses on mapping of objects from the original space to the feature space. This schema, although proven successful on a range of matching tasks, is insufficient for capturing the rich structure in the matching process of more complicated objects. In this paper, we propose a new deep architecture to more effectively model the complicated matching relations between two objects from heterogeneous domains. More specifically, we apply this model to matching tasks in natural language, e.g., finding sensible responses for a tweet, or relevant answers to a given question. This new architecture naturally combines the localness and hierarchy intrinsic to the natural language problems, and therefore greatly improves upon the state-of-the-art models.

## 1 Introduction

Many machine learning problems can be interpreted as matching two objects, e.g., images and captions in automatic captioning [11, 14], users and products in recommender systems, queries and retrieved documents in information retrieval. It is different from the usual notion of similarity since it is usually defined between objects from two different domains (e.g., texts and images), and it is usually associated with a particular purpose. The degree of matching is typically modeled as an inner-product of two representing feature vectors for objects $x$ and $y$ in a Hilbert space $\mathcal{H}$,

$$\mathsf{match}(x, y) = < \Phi_{\mathcal{Y}}(x), \Phi_{\mathcal{X}}(y) >_{\mathcal{H}} \tag{1}$$

while the modeling effort boils down to finding the mapping from the original inputs to the feature vectors. Linear models of this direction include the Partial Least Square (PLS) [19, 20], Canonical Correlation Analysis (CCA) [7], and their large margin variants [1]. In addition, there is also limited effort on finding the nonlinear mappings for that [3, 18].

In this paper, we focus on a rather difficult task of matching a given short text and candidate responses. Examples include retrieving answers for a given question and automatically commenting on a given tweet. This inner-product based schema, although proven effective on tasks like information retrieval, are often incapable for modeling the matching between complicated objects. First, representing structured objects like text as compact and meaningful vectors can be difficult; Second, inner-product cannot sufficiently take into account the complicated interaction between components within the objects, often in a rather nonlinear manner.

In this paper, we attack the problem of matching short texts from a brand new angle. Instead of representing the text objects in each domain as semantically meaningful vectors, we directly model object-object interactions with a deep architecture. This new architecture allows us to explicitly capture the natural nonlinearity and the hierarchical structure in matching two structured objects.

## 2   Model Overview

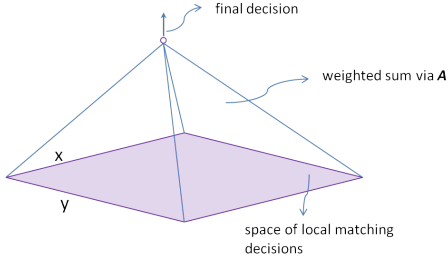

Figure 1: Architecture for linear matching.

We start with the bilinear model. Assume we can represent objects in domain $\mathcal{X}$ and $\mathcal{Y}$ with vectors $\mathbf{x} \in \mathbb{R}^{D_x}$ and $\mathbf{y} \in \mathbb{R}^{D_y}$. The bilinear matching model decides the score for any pair $(\mathbf{x}, \mathbf{y})$ as

$$\mathsf{match}(\mathbf{x}, \mathbf{y}) = \mathbf{x}^\top \mathbf{A} \mathbf{y} = \sum_{m=1}^{D_x} \sum_{n=1}^{D_y} A_{nm} x_m y_n, \quad (2)$$

with a pre-determined $\mathbf{A}$. From a different angle, each element product $x_n y_m$ in the above sum can be viewed as a micro and local decision about the matching level of $\mathbf{x}$ and $\mathbf{y}$. The outer-product matrix $\mathbf{M} = \mathbf{x}\mathbf{y}^\top$ specifies the space of element-wise interaction between objects $\mathbf{x}$ and $\mathbf{y}$. The final decision is made considering all the local decisions, while in the bilinear case $\mathsf{match}(\mathbf{x}, \mathbf{y}) = \sum_{nm} A_{nm} M_{nm}$, it simply sums all the local decisions with a weight specified by $\mathbf{A}$, as illustrated in Figure 1.

### 2.1   From Linear to Deep

This simple summarization strategy can be extended to a deep architecture to explore the nonlinearity and hierarchy in matching short texts. Unlike tasks like text classification, we need to work on a pair of text objects to be matched, which we refer to as *parallel texts*, borrowed from machine translation. This new architecture is mainly based on the following two intuitions:

**Localness:**   there is a salient local structure in the semantic space of parallel text objects to be matched, which can be roughly captured via the co-occurrence pattern of words across the objects. This localness however should not prevent two "distant" components from correlating with each other on a higher level, hence calls for the hierarchical characteristic of our model;

**Hierarchy:**   the decision making for matching has different levels of abstraction. The local decisions, capturing the interaction between semantically close words, will be combined later layer-by-layer to form the final and global decision on matching.

### 2.2   Localness

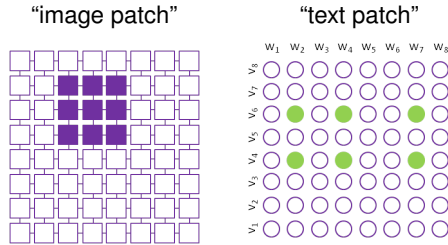

Figure 2: Image patches vs. parallel-text patches.

The localness of the text matching problem can be best described using an analogy with the patches in images, as illustrated in Figure 2. Loosely speaking, a patch for parallel texts defines the set of interacting pairs of words from the two text objects. Like the coordinate of an image patch, we can use $(\Omega_{x,p}, \Omega_{y,p})$ to specify the range of the path, with $\Omega_{x,p}$ and $\Omega_{y,p}$ each specifying a subset of terms in $\mathcal{X}$ and $\mathcal{Y}$ respectively. Like the patches of images, the patches defined here are meant to capture the segments of rich inherent structure. But unlike the naturally formed rectangular patches of images, the patches defined here do not come with a pre-given spatial continuity. It is so since in texts, the nearness of words are not naturally given as location of pixels in images, but instead needs to be discovered from the co-occurrence patterns of the matched texts. As shown later in Section 3, we actually do that with a method resembling bilingual topic modeling, which nicely captures the co-occurrence of the words within-domain and cross-domain simultaneously. The basic intuitions here are, 1) when the words co-occur frequently across the domains (e.g., `fever`—`antibiotics`), they are likely to have strong interaction in determining the matching score, and 2) when the words co-occur frequently in the same domain (e.g., {`Hawaii`,`vacation`}), they are likely to collaborate in making the matching decision. For example, modeling the matching between the word "`Hawaii`" in question (likely to be a travel-related question) and the word "`RAM`" in answer (likely an answer to a computer-related question) is probably useless, judging from their co-occurrence pattern in Question-Answer pairs. In other words, our architecture models only "local" pairwise relations on

a low level with patches, while describing the interaction between semantically distant terms on higher levels in the hierarchy.

## 2.3 Hierarchy

Once the local decisions on patches are made (most of them are NULL for a particular short text pair), they will be sent to the next layer, where the lower-level decisions are further combined to form more composite decisions, which in turn will be sent to still higher levels. This process runs until it reaches the final decision. Figure 3 gives an illustrative example on hierarchical decision making. As it shows, the local decision on patch "SIGHTSEEING IN PARIS" and "SIGHTSEEING IN BERLIN" can be combined to form a higher level decision on patch for "SIGHTSEEING", which in turn can be combined with decisions on patches like "HOTEL" and "TRANSPORTATION" to form a even higher level decision on "TRAVEL". Note that one low-level topic does not exclusively belong to a higher-level one. For example, the "WEATHER" patch may belong to higher level patches "TRAVEL" and "AGRICULTURE" at the same time.

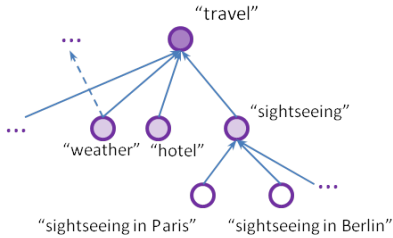

Figure 3: An example of decision hierarchy.

Quite intuitively, this decision composition mechanism is also local and varies with the "locations". For example, when combining "SIGHTSEEING IN PARIS" and "SIGHTSEEING IN BERLIN", it is more like an OR logic since it only takes one of them to be positive. A more complicated strategy is often needed in, for example, a decision on "TRAVELING", which often takes more than one element, like "SIGHTSEEING", "HOTEL", "TRANSPORTATION", or "WEATHER", but not necessarily all of them. The particular strategy taken by a local decision composition unit is fully encoded in the weights of the corresponding neuron through

$$\mathbf{s}_p(\mathbf{x}, \mathbf{y}) = f\left(\mathbf{w}_p^\top \Phi_p(\mathbf{x}, \mathbf{y})\right), \tag{3}$$

where $f$ is the active function. As stated in [12], a simple nonlinear function (such as sigmoid) with proper weights is capable of realizing basic logics such as AND and OR. Here we decide the hierarchical architecture of the decision making, but leave the exact mechanism for decision combination (encoded in the weights) to the learning algorithm later.

## 3 The Construction of Deep Architecture

The process for constructing the deep architecture for matching consists of two steps. First, we define parallel text patches with different resolutions using bilingual topic models. Second, we construct a layered directed acyclic graph (DAG) describing the hierarchy of the topics, based on which we further construct the topology of the deep neural network.

### 3.1 Topic Modeling for Parallel Texts

This step is to discover parallel text segments for meaningful co-occurrence patterns of words in both domains. Although more sophisticated methods may exist for capturing this relationship, we take an approach similar to the multi-lingual pLSI proposed in [10], and simply put the words from parallel texts together to a joint document, while using a different virtual vocabulary for each domain to avoid any mixing up. For example, the word `hotel` appearing in domain $\mathcal{X}$ is treated as a different word as `hotel` in domain $\mathcal{Y}$. For modeling tool, we use latent Dirichlet allocation (LDA) with Gibbs sampling [2] on all the training data. Notice that by using topic modeling, we allow the overlapping sets of words, which is advantageous over non-overlapping clustering of words, since we may expect some words (e.g., `hotel` and `price`) to appear in multiple segments. Table 1 gives two example parallel-topics learned from a traveling-related Question-Answer corpus (see Section 5 for more details). As we can see intuitively, in the same topic, a word in domain $\mathcal{X}$ co-occurs frequently not only with words in the same domain, but also with those in domain $\mathcal{Y}$. We fit the same corpus with $L$ topic models with decreasing resolutions[1], with the series of learned topic sets denoted as $\mathcal{H} = \{\mathcal{T}_1, \cdots, \mathcal{T}_\ell, \cdots, \mathcal{T}_L\}$, with $\ell$ indexing the topic resolution.

| Topic Label | Question | Answer |
|---|---|---|
| SPECIAL PRODUCT | local delicacy, special product snack food, quality, tasty, $\cdots$ | tofu, speciality, aroma, duck, sweet, game, cuisine sticky rice, dumpling, mushroom, traditional, $\cdots$ |
| TRANSPORTATION | route, arrangement, location arrive, train station, fare, $\cdots$ | distance, safety, spending, gateway, air ticket, pass traffic control, highway, metroplis, tunnel, $\cdots$ |

Table 1: Examples of parallel topics. Originally in Chinese, translated into English by the authors.

## 3.2 Getting Matching Architecture

With the set of topics $\mathcal{H}$, the architecture of the deep matching model can then be obtained in the following three steps. First, we trim the words (in both domains $\mathcal{X}$ and $\mathcal{Y}$) with the low probability for each topic in $\mathcal{T}_\ell \in \mathcal{H}$, and the remaining words in each topic specify a patch $p$. With a slight abuse of symbols, we still use $\mathcal{H}$ to denote the patch sets with different resolutions. Second, based on the patches specified in $\mathcal{H}$, we construct a layered DAG $\mathcal{G}$ by assigning each patch with resolution $\ell$ to a number of patches with resolution $\ell - 1$ based on the word overlapping between patches, as illustrated in Figure 4 (left panel). If a patch $p$ in layer $\ell - 1$ is assigned to patch $p'$ in layer $\ell$, we denote this relation as $p \prec p'$ [2]. Third, based on $\mathcal{G}$, we can construct the architecture of the patch-induced layers of the neural network. More specifically, each patch $p$ in layer $\ell$ will be transformed into $K_\ell$ neurons in the $(\ell-1)^{th}$ hidden layer in the neural network, and the $K_\ell$ neurons are connected to the neurons in the $\ell^{th}$ layer corresponding to patch $p'$ iff $p \prec p'$. In other words, we determine the sparsity-pattern of the weights, but leave the values of the weights to the later learning phase. Using the image analogy, the neurons corresponding to patch $p$ are referred to as *filters*. Figure 4 illustrates the process of transforming patches in layer $\ell - 1$ (specific topics) and layer $\ell$ (general topics) into two layers in neural network with $K_\ell = 2$.

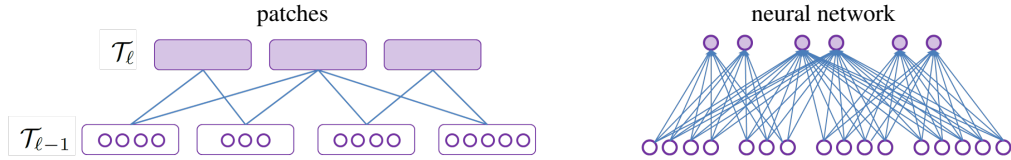

Figure 4: An illustration of constructing the deep architecture from hierarchical patches.

The overall structure is illustrated in Figure 5. The input layer is a two-dimensional interaction space, which connects to the first patch-induced layer `p-layerI` followed by the second patch-induced layer `p-layerII`. The connections to `p-layerI` and `p-layerII` have pre-specified sparsity patterns. Following `p-layerII` is a committee layer (`c-layer`), with full connections from `p-layerII`. With an input $(\mathbf{x}, \mathbf{y})$, we first get the local matching decisions on `p-layerI`, associated with patches in the interaction space. Those local decisions will be sent to the corresponding neurons in `p-layerII` to get the first round of fusion. The outputs of `p-layerII` are then sent to `c-layer` for further decision composition. Finally the logistic regression unit in the output layer summarizes the decisions on `c-layer` to get the final matching score $\mathbf{s}(\mathbf{x}, \mathbf{y})$. This architecture is referred to as DEEPMATCH in the remainder of the paper.

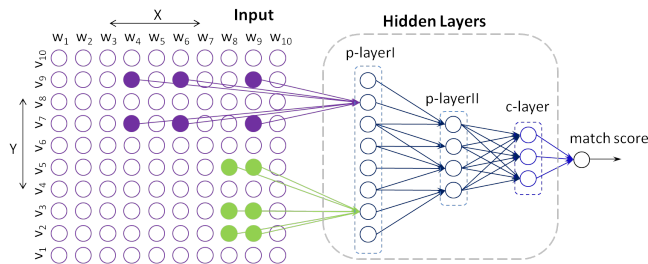

Figure 5: An illustration of the deep architecture for matching decisions.

### 3.3 Sparsity

The final constructed neural network has two types of sparsity. The first type of sparsity is enforced through architecture, since most of the connections between neurons in adjacent layers are turned off in construction. In our experiments, only about 2% of parameters are allowed to be nonzero. The second type of sparsity is from the characteristics of the texts. For most object pairs in our experiment, only a small percentage of neurons in the lower layers are active (see Section 5 for more details). This is mainly due to two factors, 1) the input parallel texts are very short (usually $< 100$ words), and 2) the patches are well designed to give a compact and sparse representation of each of the texts, as describe in Section 3.1.

To understand the second type of sparsity, let us start with the following definition:

**Definition 3.1.** An input pair $(\mathbf{x}, \mathbf{y})$ overlaps with patch $p$, iff $\mathbf{x} \cap p_x \neq \emptyset$ and $\mathbf{y} \cap p_y \neq \emptyset$, where $p_x$ and $p_y$ are respectively the word indices of patch $p$ in domain $\mathcal{X}$ and $\mathcal{Y}$.

We also define the following indicator function $\texttt{overlap}((\mathbf{x}, \mathbf{y}), p) \stackrel{\text{def}}{=} \|p_x \cap \mathbf{x}\|_0 \cdot \|p_y \cap \mathbf{y}\|_0$. The proposed architecture only allows neurons associated with patches overlapped with the input to have nonzero output. More specifically, the output of neurons associated with patch $p$ is

$$\mathbf{s}_p(\mathbf{x}, \mathbf{y}) = a_p(\mathbf{x}, \mathbf{y}) \cdot \texttt{overlap}((\mathbf{x}, \mathbf{y}), p) \qquad (4)$$

to ensure that $\mathbf{s}_p(\mathbf{x}, \mathbf{y}) \geq 0$ only when there is non-empty cross-talking of $\mathbf{x}$ and $\mathbf{y}$ within patch $p$, where $a_p(\mathbf{x}, \mathbf{y})$ is the activation of neuron before this rule is enforced. It is not hard to understand, for any input $(\mathbf{x}, \mathbf{y})$, when we track any upwards path of decisions from input to a higher level, there is nonzero matching vote until we reach a patch that contains terms from both $\mathbf{x}$ and $\mathbf{y}$. This view is particularly useful in parameter tuning with back-propagation: the supervision signal can only get down to a patch $p$ when it overlaps with input $(\mathbf{x}, \mathbf{y})$. It is easy to show from the definition, once the supervision signal stops at one patch $p$, it will not get pass $p$ and propagate to $p$'s children, even if those children have other ancestors. This indicates that when using stochastic gradient descent, the updating of weights usually only involves a very small number of neurons, and therefore can be very efficient.

### 3.4 Local Decision Models

In the hidden layers `p-layerI`, `p-layerII`, and `c-layer`, we allow two types of neurons, corresponding to two active functions: 1) linear $f_{\texttt{lin}}(t) = x$, and 2) sigmoid $f_{\texttt{sig}}(t) = (1 + e^{-t})^{-1}$. In the first layer, each patch $p$ for $(\mathbf{x}, \mathbf{y})$ takes the value of the interaction matrix $\mathbf{M}_p = \mathbf{x}_p \mathbf{y}_p^\top$, and the $k^{th}$ local decision on $p$ is given by $a_p^{(k)}(\mathbf{x}, \mathbf{y}) = f_p^{(k)} \left( \sum_{n,m} A_{p,nm}^{(k)} M_{p,nm} + b_p^{(k)} \right)$, with weight given by $\mathbf{A}^{(k)}$ and the activation function $f_p^{(k)} \in \{f_{\texttt{lin}}, f_{\texttt{sig}}\}$. With low-rank constraint on $\mathbf{A}^{(k)}$ to reduce the complexity, we essentially have

$$a_p^{(k)}(\mathbf{x}, \mathbf{y}) = f_p^{(k)} \left( \mathbf{x}_p^\top L_{x,p}^{(k)} (L_{y,p}^{(k)})^\top \mathbf{y}_p + b_p^{(k)} \right), \quad k = 1, \cdots, K_1, \qquad (5)$$

where $L_{x,p}^{(k)} \in \mathbb{R}^{|p_x| \times D_p}$, $L_{y,p}^{(k)} \in \mathbb{R}^{|p_y| \times D_p}$, with the latent dimension $D_p$. As indicated in Figure 5, the two-dimensional structure is lost after leaving the input layer, while the local structure is kept in the second patch-induced layer `p-layerII`. Basically, a neuron in layer `p-layerII` processes the low-level decisions assigned to it made in layer `p-layerI`

$$a_p^{(k)}(\mathbf{x}, \mathbf{y}) = f_p^{(k)} \left( \mathbf{w}_{p,k}^\top \Phi_p(\mathbf{x}, \mathbf{y}) \right), \quad k = 1, \cdots, K_2, \qquad (6)$$

where $\Phi_p(\mathbf{x}, \mathbf{y})$ lists all the lower-level decisions assigned to unit $p$:

$$\Phi_p(\mathbf{x}, \mathbf{y}) = [\cdots, \mathbf{s}_{p'}^{(1)}(\mathbf{x}, \mathbf{y}), \mathbf{s}_{p'}^{(2)}(\mathbf{x}, \mathbf{y}), \cdots, \mathbf{s}_{p'}^{(K_1)}(\mathbf{x}, \mathbf{y}), \cdots], \quad \forall p' \prec p, \ p' \in \mathcal{T}_1$$

which contains all the decisions on patches in layer `p-layerI` subsumed by $p$. The local decision models in the committee layer `c-layer` are the same as in `p-layerII`, except that they are fully connected to neurons in the previous layer.

## 4 Learning

We divide the parameters, denoted $\mathcal{W}$, into three sets: 1) the low-rank bilinear model for mapping from input patches to `p-layerI`, namely $L_{x,p}^{(k)}$, $L_{y,p}^{(k)}$, and offset $b_p^{(k)}$ for all $p \in \mathcal{P}$ and filter index $1 \leq k \leq K_1$, 2) the parameters for connections between patch-induced neurons, i.e., the weights

between p-layerI and p-layerII, denoted $(\mathbf{w}_p^{(k)}, b_p^{(k)})$ for associated patch $p$ and filter index $1 \le k \le K_2$, and 3) the weights for committee layer (c-layer) and after, denoted as $\mathbf{w}_c$.

We employ a discriminative training strategy with a large margin objective. Suppose that we are given the following triples $(\mathbf{x}, \mathbf{y}^+, \mathbf{y}^-)$ from the oracle, with $\mathbf{x}$ ($\in \mathcal{X}$) matched with $\mathbf{y}^+$ better than with $\mathbf{y}^-$ (both $\in \mathcal{Y}$). We have the following ranking-based loss as objective:

$$\mathcal{L}(\mathcal{W}, \mathcal{D}_{trn}) = \sum_{(\mathbf{x}_i, \mathbf{y}_i^+, \mathbf{y}_i^-) \in \mathcal{D}_{trn}} e_{\mathcal{W}}(\mathbf{x}_i, \mathbf{y}_i^+, \mathbf{y}_i^-) + R(\mathcal{W}), \tag{7}$$

where $R(\mathcal{W})$ is the regularization term, and $e_{\mathcal{W}}(\mathbf{x}_i, \mathbf{y}_i^+, \mathbf{y}_i^-)$ is the error for triple $(\mathbf{x}_i, \mathbf{y}_i^+, \mathbf{y}_i^-)$, given by the following large margin form:

$$e_i = e_{\mathcal{W}}(\mathbf{x}_i, \mathbf{y}_i^+, \mathbf{y}_i^-) = \max(0, m + \mathbf{s}(\mathbf{x}_i, \mathbf{y}_i^-) - \mathbf{s}(\mathbf{x}_i, \mathbf{y}_i^+)),$$

with $0 < m < 1$ controlling the margin in training. In the experiments, we use $m = 0.1$.

## 4.1 Back-Propagation

All three sets of parameters are updated through back-propagation (BP). The updating of the weights from hidden layers are almost the same as that for conventional Multi-layer Perceptron (MLP), with two slight differences: 1) we have a different input model and two types of activation function, and 2) we could gain some efficiency by leveraging the sparsity pattern of the neural network, but the advantage diminishes quickly after the first two layers.

This sparsity however greatly reduces the number of parameters for the first two layers, and hence the time on updating them. From Equation (4-6), the sub-gradient of $L_{x,p}^{(k)}$ w.r.t. empirical error $e$ is

$$\frac{\partial e}{\partial L_{x,p}^{(k)}} = \sum_i \Big( \frac{\partial e_i}{\partial \mathbf{s}_p^{(k)}(\mathbf{x}_i, \mathbf{y}_i^+)} \frac{\partial \mathbf{s}_p^{(k)}(\mathbf{x}_i, \mathbf{y}_i^+)}{\partial \operatorname{pot}_p^{(k)}(\mathbf{x}_i, \mathbf{y}_i^+)} \big( \mathbf{x}_{i,p}(\mathbf{y}_{i,p}^+)^\top L_{y,p}^{(k)} \big) \cdot \operatorname{overlap}\big( (\mathbf{x}_i, \mathbf{y}_i^+), p \big)$$

$$- \frac{\partial e_i}{\partial \mathbf{s}_p^{(k)}(\mathbf{x}_i, \mathbf{y}_i^-)} \frac{\partial \mathbf{s}_p^{(k)}(\mathbf{x}_i, \mathbf{y}_i^-)}{\partial \operatorname{pot}_p^{(k)}(\mathbf{x}_i, \mathbf{y}_i^-)} \big( \mathbf{x}_{i,p}(\mathbf{y}_{i,p}^-)^\top L_{y,p}^{(k)} \big) \cdot \operatorname{overlap}\big( (\mathbf{x}_i, \mathbf{y}_i^-), p \big) \Big), \tag{8}$$

where $i$ indices the training instances, and

$$\operatorname{pot}_p^{(k)}(\mathbf{x}, \mathbf{y}) = \mathbf{x}_p^\top L_{x,p}^{(k)} (L_{y,p}^{(k)})^\top \mathbf{y}_p + b_p^{(k)}$$

stands for the potential value for $\mathbf{s}_p^{(k)}$. The gradient for $L_{y,p}^{(k)}$ is given in a slightly different way. For the weights between p-layerI and p-layerII, the gradient can also benefit from the sparsity in activation.

We use stochastic sub-gradient descent with mini-batches [9], each of which consists of 50 randomly generated triples $(\mathbf{x}, \mathbf{y}^+, \mathbf{y}^-)$, where the $(\mathbf{x}, \mathbf{y}^+)$ is the original pair, and $\mathbf{y}^-$ is a randomly selected response. With this type of optimization, most of the patches in p-layerI and p-layerII get zero inputs, and therefore remain inactive by definition during the prediction as well as updating process. On the tasks we have tried, only about 2% of parameters are allowed to be nonzero for weights among the patch-induced layers. Moreover, during stochastic gradient descent, only about 5% of neurons in p-layerI and p-layerII are active on average for each training instance, indicating that the designed architecture has greatly reduced the essential capacity of the model.

## 5 Experiments

We compare our deep matching model to the inner-product based models, ranging from variants of bilinear models to nonlinear mappings for $\Phi_{\mathcal{X}}(\cdot)$ and $\Phi_{\mathcal{Y}}(\cdot)$. For bilinear models, we consider only the low-rank models with $\Phi_{\mathcal{X}}(\mathbf{x}) = P_x^\top \mathbf{x}$ and $\Phi_y(\mathbf{y}) = P_x^\top \mathbf{y}$, which gives

$$\operatorname{match}(\mathbf{x}, \mathbf{y}) = < P_x^\top \mathbf{x}, P_y^\top \mathbf{y} > = \mathbf{x}^\top P_x P_y^\top \mathbf{y}.$$

With different kinds of constraints on $P_x$ and $P_y$, we get different models. More specifically, with 1) orthnormality constraints $P_x^\top P_y = \mathbb{I}_{d \times d}$, we get partial least square (PLS) [19], and with 2) $\ell_2$ and $\ell_1$ based constraints put on rows or columns, we get Regularized Mapping to Latent Space (RMLS)

[20]. For nonlinear models, we use a modified version of the Siamese architecture [3], which uses two different neural networks for mapping objects in the two domains to the same $d$-dimensional latent space, where inner product can be used as a measure of matching and is trained with a similar large margin objective. Different from the original model in [3], we allow different parameters for mapping to handle the domain heterogeneity. Please note here that we omit the nonlinear model for shared representation [13, 18, 17] since they are essentially also inner product based models (when used for matching) and not designed to deal with short texts with large vocabulary.

## 5.1 Data Sets

We use the learned matching function for retrieving response texts $y$ for a given query text $x$, which will be ranked purely based on the matching scores. We consider the following two data sets:

Question-Answer: This data set contains around 20,000 traveling-related (Question, Answer) pairs collected from Baidu Zhidao (`zhidao.baidu.com`) and Soso Wenwen (`wenwen.soso.com`), two famous Chinese community QA Web sites. The vocabulary size is 52,315.

Weibo-Comments: This data set contains half million (Weibo, comment) pairs collected from Sina Weibo (`weibo.com`), a Chinese Twitter-like microblog service. The task is to find the appropriate responses (e.g., comments) to given Weibo posts. This task is significantly harder than the Question-Answer task since the Weibo data are usually shorter, more informal, and harder to capture with bag-of-words. The vocabulary size for tweets and comments are both $48, 724$.

On both data sets, we generate $(\mathbf{x}, \mathbf{y}^+, \mathbf{y}^-)$ triples, with $\mathbf{y}^-$ being randomly selected. The training data are randomly split into training data and testing data, and the parameters of all models (including the learned patches for DEEPMATCH) are learned on training data. The hyper parameters (e.g., the latent dimensions of low-rank models and the regularization coefficients) are tuned on a validation set (as part of the training set). We use NDCG@1 and NDCG@6 [8] on random pool with size 6 (one positive + five negative) to measure the performance of different matching models.

## 5.2 Performance Comparison

The retrieval performances of all four models are reported in Table 2. Among the two data sets, the Question-Answer data set is relatively easy, with all four matching models improve upon random guesses. As another observation, we get significant gain of performance by introducing nonlinearity in the mapping function, but all the inner-product based matching models are outperformed by the proposed DEEPMATCH with large margin on this data set. The story is slightly different on the Weibo-Response data set, which is significantly more challenging than the Q-A task in that it relies more on the content of texts and is harder to be captured by bag-of-words. This difficulty can be hardly handled by inner-product based methods, even with nonlinear mappings of SIAMESE NETWORK. In contrast, DEEPMATCH still manages to perform significantly better than all other models.

To further understand the performances of the different matching models, we also compare the generalization ability of two nonlinear models. We find that the SIAMESE NETWORK can achieve over 90% correct pairwise comparisons on training set with small regularization, but generalizes relatively poorly on the test set with all the configurations we tried. This is not surprising since SIAMESE NETWORK has the same level of parameters (varying with the number of hidden units) as DEEPMATCH. We argue that our model has better generalization property than the Siamese architecture with similar model complexity.

| | Question-Answer | | Weibo-Response | |
|---|---|---|---|---|
| | nDCG@1 | nDCG@6 | nDCG@1 | nDCG@6 |
| RANDOM GUESS | 0.167 | 0.550 | 0.167 | 0.550 |
| PLS | 0.285 | 0.662 | 0.171 | 0.587 |
| RMLS | 0.282 | 0.659 | 0.165 | 0.553 |
| SIAMESE NETWORK | 0.357 | 0.735 | 0.175 | 0.574 |
| DEEPMATCH | **0.723** | **0.856** | **0.336** | **0.665** |

Table 2: The retrieval performance of matching models on the Q-A and Weibo data sets.

### 5.3 Model Selection

We tested different variants of the current DEEPMATCH architecture, with results reported in Figure 6. There are two ways to increase the depth of the proposed method: adding patch-induced layers and committee layers. As shown in Figure 6 (left and middle panels), the performance of DEEP-MATCH stops increasing in either way when the overall depth goes beyond 6, while the training gets significantly slower with each added hidden layer. The number of neurons associated with each patch (Figure 6, right panel) follows a similar story: the performance gets flat out after the number of neurons per patch reaches 3, again with training time and memory increased significantly. As another observation about the architecture, DEEPMATCH with both linear and sigmoid activation functions in hidden layers yields slightly but consistently better performance than that with only sigmoid function. Our conjecture is that linear neurons provide shortcuts for low-level matching decision to high level composition units, and therefore facilitate the informative low-level units in determining the final matching score.

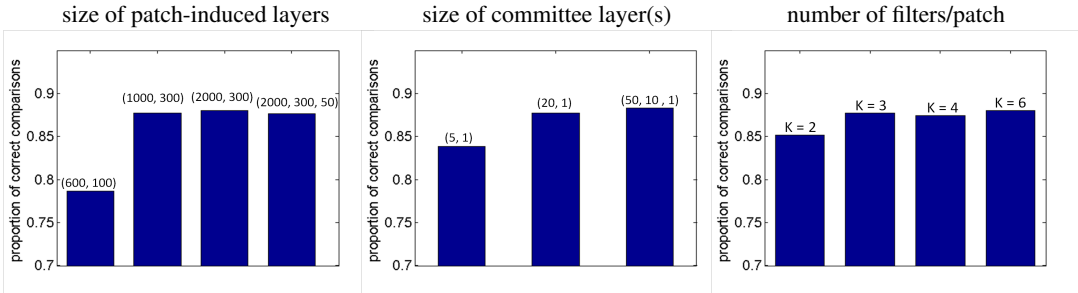

Figure 6: Choices of architecture for DEEPMATCH. For the left and middle panels, the numbers in parentheses stand for number of neurons in each layer.

## 6   Related Work

Our model is apparently a special case of the learning-to-match models, for which much effort is on designing a bilinear form [1, 19, 7]. As we discussed earlier, this kind of models cannot sufficiently model the rich and nonlinear structure of matching complicated objects. In order to introduce more modeling flexibility, there has been some works on replacing $\Phi(\cdot)$ in Equation (1) with an nonlinear mapping, e.g., with neural networks [3] or implicitly through kernelization [6]. Another similar thread of work is the recent advances of deep learning models on multi-modal input [13, 17]. It essentially finds a joint representation of inputs in two different domains, and hence can be used to predict the other side. Those deep learning models however do not give a direct matching function, and cannot handle short texts with a large vocabulary.

Our work is in a sense related to the sum-product network (SPN)[4, 5, 15], especially the work in [4] that learns the deep architecture from clustering in the feature space for the image completion task. However, it is difficult to determine a regular architecture like SPN for short texts, since the structure of the matching task for short texts is not as well-defined as that for images. We therefore adopt a more traditional MLP-like architecture in this paper.

Our work is conceptually close to the dynamic pooling algorithm recently proposed by Socher et al [16] for paraphrase identification, which is essentially a special case of matching between two homogeneous domains. Similar to our model, their proposed model also constructs a neural network on the interaction space of two objects (sentences in their case), and outputs the measure of semantic similarity between them. The major differences are three-fold, 1) their model relies on a predefined compact vectorial representation of short text, and therefore the similarity metric is not much more than summing over the local decisions, 2) the nature of dynamic pooling allows no space for exploring more complicated structure in the interaction space, and 3) we do not exploit the syntactic structure in the current model, although the proposed architecture has the flexibility for that.

## 7   Conclusion and Future Work

We proposed a novel deep architecture for matching problems, inspired partially by the long thread of work on deep learning. The proposed architecture can sufficiently explore the nonlinearity and hierarchy in the matching process, and has been empirically shown to be superior to various inner-product based matching models on real-world data sets.

## Footnotes

[1]Topic resolution is controlled mainly by the number of topics, i.e., a topic model with 100 topics is considered to be of lower resolution (or more general) than the one with 500 topics.

[2] In the assignment, we make sure each patch in layer $\ell$ is assigned to at least $m_\ell$ patches in layer $\ell - 1$.

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
