[Reviews · NeurIPS 2013]

Submitted by Assigned_Reviewer_5

This paper aims at measuring the similarity between pairs of short texts using a neural network. Each input unit is associated with a set of words from each text. Each input unit first computes a bilinear match score from its pair of word sets. Then the rest of the network is more classical. The connectivity patterns between the network units, i.e. association of terms to input units and connection between layers comes from a multi-resolution topic model.

The paper is clear and reads well. Reference to related work and application motivation are appropriate. The combination of multi-resolution connectivity structure and input bilinear match is original and seems to be nice way to address the targeted problem efficiently. I have a few suggestions to improve the paper.

** Model Overview **

I feel the analogy with image patches is not very clear. I would suggest to directly introduce the fact that you use a hierarchical topic model to build network connection, define the input layer and show how information propagate in the network, possibly with a simplified running example. This is important to understand the paper and this highlight the originality of your approach. You can take more space for that and drop gradient computations which are not necessary.

** Structure from LDA **

Getting the network structure from multiresolution LDA involves several hyperparameters: number of resolutions, number of topics per resolutions. Then getting sparse connections between words and topics as well as between topics from different resolutions involve binarizing continuous probabilities and word overlap measure. How did you pick these parameters, could you give a sense to the sensibility to this parameter. In particular, there seem to be a efficiency/effectiveness tradeoff in this binarization step.

More generally, could you show how efficient your technique is? For instance, given a computational budget on could choose to learn (i) a fully connected model (all words are used by all inputs and the rest of the net is fully connected), or (ii) your strategy with a sparsely connected model built from the topic DAG. What would be the impact on the results?

** Hyperparameter Validation **

Do you use the same rank from A for all input units? In particular, it seems that the number of words per input unit and their frequency varies.

** Typos **

pLSI -> this is confusing given that Thomas Hoffman introduced a popular model with that name as well. I would just use LSI there.
Inconsistency: NDCG@5 in the text and NEDCG@6 in the tables.
Summary: The paper is clear and reads well. Reference to related work and application motivation are appropriate. The combination of multi-resolution connectivity structure and input bilinear match is original and seems to be nice way to address the targeted problem efficiently.

Submitted by Assigned_Reviewer_6

This paper proposed a combination of bilingual topic models with a neural network based matching function.

Pros:
- interesting combination

Cons:
- overly complicated annotation and not easy to follow description
- the pipeline of running multiple LDA models with varying number of topics and then the neural network is not very elegant or principled
- I would use a hierarchical topic model instead. This seems to be a reasonable application for them.
- Generally, I would use the many deep learning based topic models and try to train the full model jointly.
- the evaluation section is very poor since the datasets are new and not available to the public and the random baseline scores quite highly too.

are the patches just words from the different topics?

the figures such as fig. 5 could be clearer
the notation is unnecessarily complicated and not easy to follow, e.g. a_p(x,y) is used and then only defined several paragraphs later.

if possible, please find a native speaker to proof read, there are a couple of grammar mistakes,
"the remained words", "increasing with either way"

the claim that deep learning models do not give matching functions and cannot handle short texts with large vocabularies is just not true. You even cite the paper by Socher et al. that deals with large vocabularies and single sentences for paraphrase detection.


what is the speed of your method?
Summary: This paper proposed a combination of bilingual topic models with a neural network based matching function.

The results are hard to replicate since the data is not available and some claims are not true but there are some interesting ideas in there.

Submitted by Assigned_Reviewer_7

This paper proposes an approach to matching short texts based on using topic
modeling to identify the common word co-occurrence patterns within pairs of
matched documents. Topic modeling is peformed several times using fewer and
fewer topics, producing a hierarchy over topics and thus word co-occurrence
patterns. The heirarchy is then used to define the structure of a sparsely
connected neural network, taking advantage of "semantic locality". The model is
then trained to score correct matches above incorrect ones.

The approach is novel and potentially interesting, but is fairly complicated
and not very clearly described in the paper. While the analogy between image
patches and word groups of co-occurring words is probably worth mentioning,
using the image patch terminology throughout the paper is more confusing than
helpful.

The experimental section is the weakest point of the paper. The authors claim
that their model performs so well because it is hierarchical and nonlinear, but
the experimental results do not provide direct evidence for these claims. To
substantiate these claims, DeepMatch should be compared to a linear version of
itself (with all sigmoid units replaced by linear ones) as well as to a
non-hierarchical version of itself (using only one resolution of topics).
Without these comparisons there are several confounding factors that could
potentially explain the superior results obtained by DeepMatch. As for using
existing baselines, it makes more sense to compare to Supervised Semantic
Indexing or Polynomial Semantic Indexing instead of RMLS, as those methods are
trained using the same margin-based criterion as DeepMatch. The choice of the
datasets is also unfortunate, as they appear to be non-standard and
proprietary, which makes the reported scores difficult to interpret and the
results impossible to reproduce. The authors also do not report several
crucial details such as the latent dimensionality of the models, the number of
topics used, the number hidden units (sigmoid vs. linear), regularization
parameters etc. The description of the hyperparameter selection procedure on
lines 345-48 is a cause for concern as it seems to suggest that the
hyperparemeters where selected on a subset of the training set, which is likely
to result in overfitting. The big gap in performance between DeepMatch and the
siamese network makes me wonder whether the latter was properly regularized.
The mention of "zero regularization" for that model on line 364 seems to
suggest otherwise. Given the relatively small size of the datasets for the
vocabulary sizes used, proper model regularization is absolutely essential for
a fair comparison.

The gradients given by Eqs. (8) and (9) are not quite right as they reference
y_i instead of y+_i and y-_i. It is unclear what the potential value (pot^k_p)
is. Is it the total input to a unit?

It appears that DeepMatch used a reduced vocabulary ("trim the words" on line
172). Is this correct and if so, was the same vocabulary reduction performed
for all models?

How exactly are the higher-resolution topics are assigned to the lower-resolution
ones?

Were the comparison accuracy scores on Figure 6 computed on the training data or
on the test data? What are the "correct comparisons" shown on the figure?

The paper repeatedly uses "localness" instead of the more appropriate "locality".
Summary: The idea is potentially interesting, but the write up is unclear and the experimental
evaluation not convincing.

Submitted by Assigned_Reviewer_8

This paper proposes a new deep architecture for matching texts from two categories (such as questions and answers). The main originality consists in defining the architecture of the neural network using hierarchical topic models, trained jointly on both text categories. The topic models are used to define word patches of different granularities as well as hierarchical connections between them. These patches and connections then serve as to define the connectivity pattern of a neural network (later trained by backprop+sgd).


The paper is fairly written and quite easy to follow.

The main idea of the paper is neat and original I think. Relying on LDA to define the text patches is very nice. So is it for deriving the connectivity pattern of the network.

The main weakness of this paper is the experimental part. The main assumptions underlying the introduced framework are:
(1) models for dealing with text should take its inherent structure into account;
(2) such structure does not necessarily follow the ordering of words in the text, nor does it follow any kind of parse tree.

The main originality of the paper concerns (2) since many previous approaches agree with (1). And here is the problem: to justify claim (2) (which I am inclined to accept), one should compare with previous approaches using text structure in the model. But there is no such comparison (only variants using bag-of-words). Some interesting comparisons could be:

* siamese networks with ngrams or with parse tree features (from syntactic or dependency parses). Easy to do and already considers some structure.
* non-linear neural networks that uses the word ordering as structure (such as in Collobert et al. JMLR 2011).
* the RNN of Socher et al.
* and how about using directly the hierarchical topic models to match the short texts?
Summary: This paper introduces a new nice framework for "learning" the structure of neural networks dealing with text as input. Unfortunately, the experiments do not allow to completely assess the efficiency of this approach.
Author Feedback

Author rebuttal: We thank all 4 reviewers for their insightful comments. We are working towards making the data-set publically available in near future.

To Reviewer 5:
1.We used the same rank for all the local bi-linear models
2.We agree with the reviewer that a comparison between different models with fixed “computational budget” would be quite meaningful

To Reviewer 6:
1.We agree that a hierarchical LDA is a more elegant choice for finding the architecture of the model, but other heuristics in the constructing process (eg. assigning at least k more specific clusters to a more general one) will probably diminish the elegance of HLDA.
2.To answer the question (“are the patches just words from the different topics? ”): each patch depicts the interaction between words from two domains in a concatenated topic in the ``bilingual” topic model.
3.To answer the comment (“the claim that deep learning models do not give matching functions and cannot handle short texts with large vocabularies is just not true. You even cite the paper by Socher et al. that deals with large vocabularies and single sentences for paraphrase detection. ? ”): Socher et al model (ref[16]) is based on word embedding and therefore avoids the difficulty brought by the large vocabulary , but its setting only applies to paraphrase detection.
4.It took about 10 hours training with 100,000 triples on a PC.

To Reviewer 7:
1.We tried several settings of regularization for the Siamese network, and actually reported the best one (which is not significantly better than the un-regularized one). Our argument with “zero regularization” is merely to point out that the inferior performance of the Siamese network is not due to its lack of model capacity.
2.Yes, Eq (8) & (9) only give the gradient from a term in the error function, and therefore incomplete.
3.We thank the reviewer for pointing out the work of Supervised Semantic Indexing. We didn’t include any comparison to it since Siamese network is a nonlinear version of it (with same margin-based objective), but we will include the empirical comparison to it in the future version.
4.In Section 5.3, we already reported the performance with several variants of the architectures, which indicates that a shallower architecture performs worse. We omitted several primitive ones (eg. the linear combination of local matchings) but will consider including them in the further version
5.Yes, we use the same reduced vocabulary for all models.
6.Figure 6 reports the performance on test data


To Reviewer 8:
1.We agree with the reviewer that some comparison to model with natural language structure would be interesting. However, the current models (e.g Socher et al’s RNN & the C&W’s RENNA) cannot be directly applied to the matching cases.
2.A hierarchical topic models would give a multi-resolution representation of the text, which can be used for learning a matching model. We may consider including experiments like that in the future version of the paper.